# Automated Variational Inference
# for Gaussian Process Models

**Trung V. Nguyen**
ANU & NICTA
VanTrung.Nguyen@nicta.com.au

**Edwin V. Bonilla**
The University of New South Wales
e.bonilla@unsw.edu.au

## Abstract

We develop an automated variational method for approximate inference in Gaussian process (GP) models whose posteriors are often intractable. Using a mixture of Gaussians as the variational distribution, we show that (i) the variational objective and its gradients can be approximated efficiently via sampling from univariate Gaussian distributions and (ii) the gradients wrt the GP hyperparameters can be obtained analytically regardless of the model likelihood. We further propose two instances of the variational distribution whose covariance matrices can be parametrized linearly in the number of observations. These results allow gradient-based optimization to be done efficiently in a black-box manner. Our approach is thoroughly verified on five models using six benchmark datasets, performing as well as the exact or hard-coded implementations while running orders of magnitude faster than the alternative MCMC sampling approaches. Our method can be a valuable tool for practitioners and researchers to investigate new models with minimal effort in deriving model-specific inference algorithms.

## 1 Introduction

Gaussian processes (GPs, [1]) are a popular choice in practical Bayesian non-parametric modeling. The most straightforward application of GPs is the standard regression model with Gaussian likelihood, for which the posterior can be computed in closed form. However, analytical tractability is no longer possible when having non-Gaussian likelihoods, and inference must be carried out via approximate methods, among which Markov chain Monte Carlo (MCMC, see e.g. [2]) and variational inference [3] are arguably the two techniques most widely used.

MCMC algorithms provide a flexible framework for sampling from complex posterior distributions of probabilistic models. However, their generality comes at the expense of very high computational cost as well as cumbersome convergence analysis. Furthermore, methods such as Gibbs sampling may perform poorly when there are strong dependencies among the variables of interest. Other algorithms such as the elliptical slice sampling (ESS) developed in [4] are more effective at drawing samples from strongly correlated Gaussians. Nevertheless, while improving upon generic MCMC methods, the sampling cost of ESS remains a major challenge for practical usages.

Alternative to MCMC is the deterministic approximation approach via variational inference, which has been used in numerous applications with some empirical success ( see e.g. [5, 6, 7, 8, 9, 10, 11]). The main insight from variational methods is that optimizing is generally easier than integrating. Indeed, they approximate a posterior by optimizing a lower bound of the marginal likelihood, the so-called evidence lower bound (ELBO). While variational inference can be considerably faster than MCMC, it lacks MCMC's broader applicability as it requires derivations of the ELBO and its gradients on a model-by-model basis.

This paper develops an *automated variational inference* technique for GP models that not only reduces the overhead of the tedious mathematical derivations inherent to variational methods but also

allows their application to a wide range of problems. In particular, we consider Gaussian process models that satisfy the following properties: (i) factorization across latent functions and (ii) factorization across observations. The former assumes that, when there is more than one latent function, they are generated from independent GPs. The latter assumes that, given the latent functions, the observations are conditionally independent. Existing GP models, such as regression [1], binary and multi-class classification [6, 12], warped GPs [13], log Gaussian Cox process [14], and multi-output regression [15], all fall into this class of models. We note, however, that our approach goes beyond standard settings for which elaborate learning machinery has been developed, as we only require access to the likelihood function in a black-box manner.

Our automated deterministic inference method uses a mixture of Gaussians as the approximating posterior distribution and exploits the decomposition of the ELBO into a KL divergence term and an expected log likelihood term. In particular, we derive an analytical lower bound for the KL term; and we show that the expected log likelihood term and its gradients can be computed efficiently by sampling from univariate Gaussian distributions, without explicitly requiring gradients of the likelihood. Furthermore, we optimize the GP hyperparameters within the same variational framework by using their analytical gradients, irrespective of the specifics of the likelihood models.

Additionally, we exploit the efficient parametrization of the covariance matrices in the models, which is linear in the number of observations, along with variance-reduction techniques in order to provide an automated inference framework that is useful in practice. We verify the effectiveness of our method with extensive experiments on 5 different GP settings using 6 benchmark datasets. We show that our approach performs as well as exact GPs or hard-coded deterministic inference implementations, and that it can be up to several orders of magnitude faster than state-of-the-art MCMC approaches.

**Related work**

Black box variational inference (BBVI, [16]) has recently been developed for general latent variable models. Due to this generality, it under-utilizes the rich amount of information available in GP models that we previously discussed. For example, BBVI approximates the KL term of the ELBO, but this is computed analytically in our method. A clear disadvantage of BBVI is that it does not provide an analytical or practical way of learning the covariance hyperparameters of GPs – in fact, these are set to fixed values. In principle, these values can be learned in BBVI using stochastic optimization, but experimentally, we have found this to be problematic, ineffectual, and time-consuming. In contrast, our method optimizes the hyperparameters using their exact gradients.

An approach more closely related to ours is in [17], which investigates variational inference for GP models with one latent function and factorial likelihood. Their main result is an efficient parametrization when using a standard variational Gaussian distribution. Our method is more general in that it allows multiple latent functions, hence being applicable to settings such as multi-class classification and multi-output regression. Furthermore, our variational distribution is a mixture of Gaussians, with the full Gaussian distribution being a particular case. Another recent approach to deterministic approximate inference is the Integrated Nested Laplace Approximation (INLA, [18]). INLA uses numerical integration to approximate the marginal likelihood, which makes it unsuitable for GP models that contain a large number of hyperparameters.

## 2 A family of GP models

We consider supervised learning problems with a dataset of $N$ training inputs $\mathbf{x} = \{\mathbf{x}_n\}_{n=1}^N$ and their corresponding targets $\mathbf{y} = \{\mathbf{y}_n\}_{n=1}^N$. The mapping from inputs to outputs is established via $Q$ underlying latent functions, and our objective is to reason about these latent functions from the observed data. We specify a class of GP models for which the priors and the likelihoods have the following structure:

$$p(\mathbf{f}|\boldsymbol{\theta}_0) = \prod_{j=1}^Q p(\mathbf{f}_{\bullet j}|\boldsymbol{\theta}_0) = \prod_{j=1}^Q \mathcal{N}(\mathbf{f}_{\bullet j}; \mathbf{0}, \mathbf{K}_j), \tag{1}$$

$$p(\mathbf{y}|\mathbf{f}, \boldsymbol{\theta}_1) = \prod_{n=1}^N p(\mathbf{y}_n|\mathbf{f}_{n\bullet}, \boldsymbol{\theta}_1), \tag{2}$$

where $\mathbf{f}$ is the set of all latent function values; $\mathbf{f}_{\bullet j} = \{f_j(\mathbf{x}_n)\}_{n=1}^N$ denotes the values of the latent function $j$; $\mathbf{f}_{n\bullet} = \{f_j(\mathbf{x}_n)\}_{j=1}^Q$ is the set of latent function values which $\mathbf{y}_n$ depends upon; $\mathbf{K}_j$ is the covariance matrix evaluated at every pair of inputs induced by the covariance function $k_j(\cdot, \cdot)$; and $\boldsymbol{\theta}_0$ and $\boldsymbol{\theta}_1$ are covariance hyperparameters and likelihood parameters, respectively.

In other words, the class of models specified by Equations (1) and (2) satisfy the following two criteria: (a) factorization of the prior over the latent functions and (b) factorization of the conditional likelihood over the observations. Existing GP models including GP regression [1], binary classification [6, 12], warped GPs [13], log Gaussian Cox processes [14], multi-class classification [12], and multi-output regression [15] all belong to this family of models.

## 3  Automated variational inference for GP models

This section describes our automated inference framework for posterior inference of the latent functions for the given family of models. Apart from Equations (1) and (2), we only require access to the likelihood function in a black-box manner, i.e. specific knowledge of its shape or its gradient is not needed. Posterior inference for general (non-Gaussian) likelihoods is analytically intractable.

We build our posterior approximation framework upon variational inference principles. This entails positing a tractable family of distributions and finding the member of the family that is "closest" to the true posterior in terms of their KL divergence. Herein we choose the family of mixture of Gaussians (MoG) with $K$ components, defined as

$$q(\mathbf{f}|\boldsymbol{\lambda}) = \frac{1}{K}\sum_{k=1}^K q_k(\mathbf{f}|\mathbf{m}_k, \mathbf{S}_k) = \frac{1}{K}\sum_{k=1}^K\prod_{j=1}^Q \mathcal{N}(\mathbf{f}_{\bullet j}; \mathbf{m}_{kj}, \mathbf{S}_{kj}), \quad \boldsymbol{\lambda} = \{\mathbf{m}_{kj}, \mathbf{S}_{kj}\}, \qquad (3)$$

where $q_k(\mathbf{f}|\mathbf{m}_k, \mathbf{S}_k)$ is the component $k$ with variational parameters $\mathbf{m}_k = \{\mathbf{m}_{kj}\}_{j=1}^Q$ and $\mathbf{S}_k = \{\mathbf{S}_{kj}\}_{j=1}^Q$. Less general MoG with *isotropic* covariances have been used with variational inference in [7, 19]. Note that within each component, the posteriors over the latent functions are independent.

Minimizing the divergence $\text{KL}[q(\mathbf{f}|\boldsymbol{\lambda})||p(\mathbf{f}|\mathbf{y})]$ is equivalent to maximizing the evidence lower bound (ELBO) given by:

$$\log p(\mathbf{y}) \geq \underbrace{\mathbb{E}_q[-\log q(\mathbf{f}|\boldsymbol{\lambda})] + \mathbb{E}_q[\log p(\mathbf{f})]}_{-\text{KL}[q(\mathbf{f}|\boldsymbol{\lambda})||p(\mathbf{f})]} + \frac{1}{K}\sum_{k=1}^K\mathbb{E}_{q_k}[\log p(\mathbf{y}|\mathbf{f})] \triangleq \mathcal{L}. \qquad (4)$$

Observe that the KL term in Equation (4) does not depend on the likelihood. The remaining term, called the *expected log likelihood* (ELL), is the only contribution of the likelihood to the ELBO. We can thus address the technical difficulties regarding each component and their derivatives separately using different approaches. In particular, we can obtain a lower bound of the first term (KL) and approximate the second term (ELL) via sampling. Due to the limited space, we only show the main results and refer the reader to the supplementary material for derivation details.

### 3.1  A lower bound of $-\mathbf{KL}[q(\mathbf{f}|\boldsymbol{\lambda})||p(\mathbf{f})]$

The first component of the KL divergence term is the entropy of a Gaussian mixture which is not analytically tractable. However, a lower bound of this entropy can be obtained using Jensen's inequality (see e.g. [20]) giving:

$$\mathbb{E}_q[-\log q(\mathbf{f}|\boldsymbol{\lambda})] \geq -\sum_{k=1}^K\frac{1}{K}\log\sum_{l=1}^K\frac{1}{K}\mathcal{N}(\mathbf{m}_k; \mathbf{m}_l, \mathbf{S}_k + \mathbf{S}_l). \qquad (5)$$

The second component of the KL term is a negative cross-entropy between a Gaussian mixture and a Gaussian, which can be computed analytically giving:

$$\mathbb{E}_q[\log p(\mathbf{f})] = -\frac{1}{2K}\sum_{k=1}^K\sum_{j=1}^Q\left[N\log 2\pi + \log|\mathbf{K}_j| + \mathbf{m}_{kj}^T\mathbf{K}_j^{-1}\mathbf{m}_{kj} + \text{tr}\left(\mathbf{K}_j^{-1}\mathbf{S}_{kj}\right)\right]. \qquad (6)$$

The gradients of the two terms in Equations (5) and (6) wrt the variational parameters can be computed analytically and are given in the supplementary material.

## 3.2 An approximation to the expected log likelihood (ELL)

It is clear from Equation (4) that the ELL can be obtained via the ELLs of the individual mixture components $\mathbb{E}_{q_k}[\log p(\mathbf{y}|\mathbf{f})]$. Due to the factorial assumption of $p(\mathbf{y}|\mathbf{f})$, the expectation becomes:

$$\mathbb{E}_{q_k}[\log p(\mathbf{y}|\mathbf{f})] = \sum_{n=1}^{N} \mathbb{E}_{q_{k(n)}}[\log p(\mathbf{y}_n|\mathbf{f}_{n\bullet})], \qquad (7)$$

where $q_{k(n)} = q_{k(n)}(\mathbf{f}_{n\bullet}|\boldsymbol{\lambda}_{k(n)})$ is the marginal posterior with variational parameters $\boldsymbol{\lambda}_{k(n)}$ that correspond to $\mathbf{f}_{n\bullet}$. The gradients of these individual ELL terms wrt the variational parameters $\boldsymbol{\lambda}_{k(n)}$ are given by:

$$\nabla_{\boldsymbol{\lambda}_{k(n)}}\mathbb{E}_{q_{k(n)}}[\log p(\mathbf{y}_n|\mathbf{f}_{n\bullet})] = \mathbb{E}_{q_{k(n)}}\nabla_{\boldsymbol{\lambda}_{k(n)}}\log q_{k(n)}(\mathbf{f}_{n\bullet}|\boldsymbol{\lambda}_{k(n)})\log p(\mathbf{y}_n|\mathbf{f}_{n\bullet}). \qquad (8)$$

Using Equations (7) and (8) we establish the following theorem regarding the computation of the ELL and its gradients.

**Theorem 1.** *The expected log likelihood and its gradients can be approximated using samples from univariate Gaussian distributions.*

The proof is in Section 1 of the supplementary material. A less general result, for the case of one latent function and the variational Gaussian posterior, was obtained in [17] using a different derivation. Note that when $Q > 1$, $q_{k(n)}$ is not a univariate marginal. Nevertheless, it has a diagonal covariance matrix due to the factorization of the latent posteriors so the theorem still holds.

## 3.3 Learning of the variational parameters and other model parameters

In order to learn the parameters of the model we use gradient-based optimization of the ELBO. For this we require the gradients of the ELBO wrt all model parameters.

**Variational parameters.** The noisy gradients of the ELBO w.r.t. the variational means $\mathbf{m}_{k(n)}$ and variances $\mathbf{S}_{k(n)}$ corresponding to data point $n$ are given by:

$$\hat{\nabla}_{\mathbf{m}_{k(n)}}\mathcal{L} \approx \nabla_{\mathbf{m}_{k(n)}}\mathcal{L}_{\text{ent}} + \nabla_{\mathbf{m}_{k(n)}}\mathcal{L}_{\text{cross}} + \frac{1}{KS}\mathbf{s}_{k(n)}^{-1} \circ \sum_{i=1}^{S}(\mathbf{f}_{n\bullet}^i - \mathbf{m}_{k(n)})\log p(\mathbf{y}_n|\mathbf{f}_{n\bullet}^i), \qquad (9)$$

$$\hat{\nabla}_{\mathbf{S}_{k(n)}}\mathcal{L} \approx \nabla_{\mathbf{S}_{k(n)}}\mathcal{L}_{\text{ent}} + \nabla_{\mathbf{S}_{k(n)}}\mathcal{L}_{\text{cross}}$$
$$+ \frac{1}{2KS}\text{dg}\sum_{i=1}^{S}\left(\mathbf{s}_{k(n)}^{-1} \circ \mathbf{s}_{k(n)}^{-1} \circ (\mathbf{f}_{n\bullet}^i - \mathbf{m}_{k(n)}) \circ (\mathbf{f}_{n\bullet}^i - \mathbf{m}_{k(n)}) - \mathbf{s}_{k(n)}^{-1}\right)\log p(\mathbf{y}_n|\mathbf{f}_{n\bullet}^i) \quad (10)$$

where $\circ$ is the entrywise Hadamard product; $\{\mathbf{f}_{n\bullet}^i\}_{i=1}^S$ are samples from $q_{k(n)}(\mathbf{f}_{n\bullet}|\mathbf{m}_{k(n)}, \mathbf{s}_{k(n)})$; $\mathbf{s}_{k(n)}$ is the diagonal of $\mathbf{S}_{k(n)}$ and $\mathbf{s}_{k(n)}^{-1}$ is the element-wise inverse of $\mathbf{s}_{k(n)}$; dg turns a vector to a diagonal matrix; and $\mathcal{L}_{\text{ent}} = \mathbb{E}_q[-\log q(\mathbf{f}|\boldsymbol{\lambda})]$ and $\mathcal{L}_{\text{cross}} = \mathbb{E}_q[\log p(\mathbf{f})]$ are given by Equations (5) and (6). The control variates technique described in [16] is also used to further reduce the variance of these estimators.

**Covariance hyperparameters.** The ELBO in Equation (4) reveals a remarkable property: the hyperparameters depend only on the negative cross-entropy term $\mathbb{E}_q[\log p(\mathbf{f})]$ whose *exact* expression was derived in Equation (6). This has a significant practical implication: despite using black-box inference, the hyperparameters are optimized wrt the true evidence lower bound (given fixed variational parameters). This is an additional and crucial advantage of our automated inference method over other generic inference techniques [16] that seem incapable of hyperparameter learning, in part because there are not yet techniques for reducing the variance of the gradient estimators. The gradient of the ELBO wrt any hyperparameter $\theta$ of the $j$-th covariance function is given by:

$$\nabla_{\theta}\mathcal{L} = -\frac{1}{2K}\sum_{k=1}^{K}\text{tr}\left(\mathbf{K}_j^{-1}\nabla_{\theta}\mathbf{K}_j - \mathbf{K}_j^{-1}\nabla_{\theta}\mathbf{K}_j\mathbf{K}_j^{-1}(\mathbf{m}_{kj}\mathbf{m}_{kj}^T + \mathbf{S}_j)\right). \qquad (11)$$

**Likelihood parameters**    The noisy gradients w.r.t. the likelihood parameters can also be estimated via samples from univariate marginals:

$$\nabla_{\boldsymbol{\theta}_1} \mathcal{L} \approx \frac{1}{KS} \sum_{k=1}^{K} \sum_{n=1}^{N} \sum_{i=1}^{S} \nabla_{\boldsymbol{\theta}_1} \log p(\mathbf{y}_n | \mathbf{f}_{(n)}^{k,i}, \boldsymbol{\theta}_1), \text{ where } \mathbf{f}_{(n)}^{k,i} \sim q_{k(n)}(\mathbf{f}_{n\bullet} | \mathbf{m}_{k(n)}, \mathbf{s}_{k(n)}). \quad (12)$$

## 3.4   Practical variational distributions

The gradients from the previous section may be used for automated variational inference for GP models. However, the mixture of Gaussians (MoG) requires $\mathcal{O}(N^2)$ variational parameters for each covariance matrix, i.e. we need to estimate a total of $\mathcal{O}(QKN^2)$ parameters. This causes difficulties for learning when these parameters are optimized simultaneously. This section introduces two special members of the MoG family that improve the practical tractability of our inference framework.

**Full Gaussian posterior.**    This instance is the mixture with only 1 component and is thus a Gaussian distribution. Its covariance matrix has block diagonal structure, where each block is a full covariance corresponding to that of a single latent function posterior. We thus refer to it as the full Gaussian posterior. As stated in the following theorem, full Gaussian posteriors can still be estimated efficiently in our variational framework.

**Theorem 2.** *Only $\mathcal{O}(QN)$ variational parameters are required to parametrize the latent posteriors with full covariance structure.*

The proof is given Section 2 of the supplementary material. This result has been stated previously (see e.g. [6, 7, 17]) but for specific models that belong to the class of GP models considered here.

**Mixture of diagonal Gaussians posterior.**    Our second practical variational posterior is a Gaussian mixture with diagonal covariances, yielding two immediate benefits. Firstly, only $\mathcal{O}(QN)$ parameters are required for each mixture component. Secondly, computation is more efficient as inverting a diagonal covariance can be done in linear time. Furthermore, as a result of the following theorem, optimization will typically converge faster when using a mixture of diagonal Gaussians.

**Theorem 3.** *The estimator of the gradients wrt the variational parameters using the mixture of diagonal Gaussians has a lower variance than the full Gaussian posterior's.*

The proof is in Section 3 of the supplementary material and is based on the Rao-Blackwellization technique [21]. We note that this result is different to that in [16]. In particular, our variational distribution is a mixture, thus multi-modal. The theorem is only made possible due to the analytical tractability of the KL term in the ELBO.

Given the noisy gradients, we use off-the-shelf, gradient-based optimizers, such as conjugate gradient, to learn the model parameters. Note that stochastic optimization may also be used, but it may require significant time and effort in tuning the learning rates.

## 3.5   Prediction

Given the MoG posterior, the predictive distribution for new test points $\mathbf{x}_*$ is given by:

$$p(\mathbf{Y}_* | \mathbf{x}_*) = \frac{1}{K} \sum_{k=1}^{K} \int p(\mathbf{Y}_* | \mathbf{f}_*) \int p(\mathbf{f}_* | \mathbf{f}) q_k(\mathbf{f}) \mathrm{d}\mathbf{f} \mathrm{d}\mathbf{f}_*. \quad (13)$$

The inner integral is the predictive distribution of the latent values $f_*$ and it is a Gaussian since both $q_k(\mathbf{f})$ and $p(\mathbf{f}_* | \mathbf{f})$ are Gaussian. The probability of the test points taking values $\mathbf{y}_*$ (e.g. in classification) can thus be readily estimated via Monte Carlo sampling. The predictive means and variances of a MoG can be obtained from that of the individual mixture components as described in Section 6 of the supplementary material.

Table 1: Datasets, their statistics, and the corresponding likelihood functions and models used in the experiments, where $N_{train}$, $N_{test}$, and $D$ are the training size, testing size, and the input dimension, respectively. See text for detailed description of the models.

| Dataset | $N_{train}$ | $N_{test}$ | $D$ | Likelihood $p(y|f)$ | Model |
|---|---|---|---|---|---|
| Mining disasters | 811 | 0 | 1 | $\lambda^y \exp(-\lambda)/y!$ | Log Gausian Cox process |
| Boston housing | 300 | 206 | 13 | $\mathcal{N}(y; f, \sigma^2)$ | Standard regression |
| Creep | 800 | 1266 | 30 | $\nabla_y t(y)\mathcal{N}(t(y); f, \sigma^2)$ | Warped Gaussian processes |
| Abalone | 1000 | 3177 | 8 | same as above | Warped Gaussian processes |
| Breast cancer | 300 | 383 | 9 | $1/(1 + \exp(-f))$ | Binary classification |
| USPS | 1233 | 1232 | 256 | $\exp(f_c)/\sum_{i=1} \exp(f_i)$ | Multi-class classification |

# 4 Experiments

We perform experiments with five GP models: standard regression [1], warped GPs [13], binary classification [6, 12], multi-class classification [12], and log Gaussian Cox processes [14] on six datasets (see Table 1) and repeat the experiments five times using different data subsets.

**Experimental settings.** The squared exponential covariance function with automatic relevance determination (see Ch. 4 in [1]) is used with the GP regression and warped GPs. The isotropic covariance is used with all other models. The noisy gradients of the ELBO are approximated with 2000 samples and 200 samples are used with control variates to reduce the variance of the gradient estimators. The model parameters (variational, covariance hyperparameters and likelihood parameters) are learned by iteratively optimizing one set while fixing the others until convergence, which is determined when changes are less than $1e$-5 for the ELBO or $1e$-3 for the variational parameters.

**Evaluation metrics.** To assess the predictive accuracy, we use the standardized squared error (SSE) for the regression tasks and the classification error rates for the classification tasks. The negative log predictive density (NLPD) is also used to evaluate the confidence of the prediction. For all of the metrics, smaller figures are better.

**Notations.** We call our method AGP and use AGP-FULL, AGP-MIX and AGP-MIX2 when using the full Gaussian and the mixture of diagonal Gaussians with 1 and 2 components, respectively. Details of these two posteriors were given in Section 3.4. On the plots, we use the shorter notations, FULL, MIX, and MIX2 due to the limited space.

**Reading the box plots.** We used box plots to give a more complete picture of the predictive performance. Each plot corresponds to the distribution of a particular metric evaluated at all test points for a given task. The edges of a box are the $q_1 = 25$th and $q_3 = 75$th percentiles and the central mark is the median. The dotted line marks the limit of extreme points that are greater than the $97.5$th percentile. The whiskers enclose the points in the range $(q_1 - 1.5(q_3 - q_1), q_3 + 1.5(q_3 - q_1))$, which amounts to approximately $\pm 2.7\sigma$ if the data is normally distributed. The points outside the whiskers and below the dotted line are outliers and are plotted individually.

## 4.1 Standard regression

First we consider the standard Gaussian process regression for which the predictive distribution can be computed analytically. We compare with this exact inference method (GPR) using the Boston housing dataset [22]. The results in Figure 1 show that AGP-FULL achieves nearly identical performance as GPR. This is expected as the analytical posterior is a full Gaussian. AGP-MIX and AGP-MIX2 also give comparable performance in terms of the median SSE and NLPD.

## 4.2 Warped Gaussian processes (WGP)

The WGP allows for non-Gaussian processes and non-Gaussian noises. The likelihood for each target $y_n$ is attained by warping it through a nonlinear monotonic transformation $t(y)$ giving $p(y_n|f_n) = \nabla_{y_n} t(y_n)\mathcal{N}(t(y_n)|f_n, \sigma^2)$. We used the same neural net style transformation as in [13]. We fixed the warp parameters and used the same procedure for making analytical approximations to the predicted means and variances for all methods.

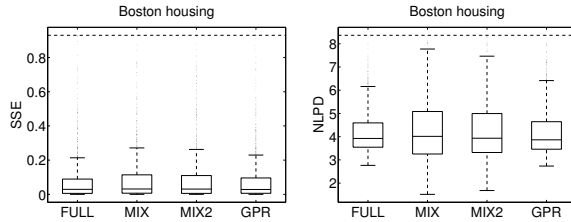

Figure 1: The distributions of SSE and NLPD of all methods on the regression task. Compared to the exact inference method GPR, the performance of AGP-FULL is identical while that of AGP-MIX and AGP-MIX2 are comparable.

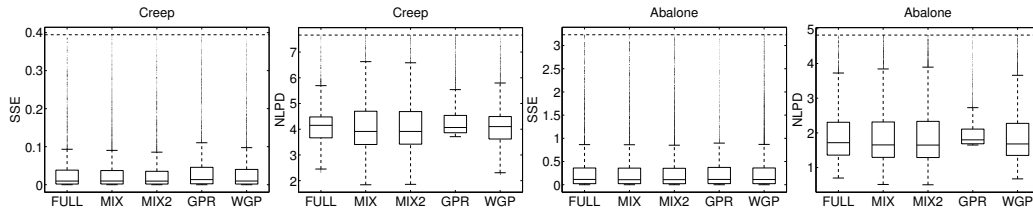

Figure 2: The distributions of SSE and NLPD of all methods on the regression task with warped GPs. The AGP methods (FULL, MIX and MIX2) give comparable performance to exact inference with WGP and slightly outperform GPR which has narrower ranges of predictive variances.

We compare with the exact implementation of [13] and the standard GP regression (GPR) on the Creep [23] and Abalone [22] datasets. The results in Figure 2 show that the AGP methods give comparable performance to the exact method WGP and slightly outperform GPR. The prediction by GPR exhibits characteristically narrower ranges of predictive variances which can be attributed to its Gaussian noise assumption.

### 4.3 Classification

For binary classification, we use the logistic likelihood and experiment with the Wisconsin breast cancer dataset [22]. We compare with the variational bounds (VBO) and the expectation propagation (EP) methods. Details of VBO and EP can be found in [6]. All methods use the same analytical approximations when making prediction.

For multi-class classification, we use the softmax likelihood and experiment with a subset of the USPS dataset [1] containing the digits 4, 7, and 9. We compare with a variational inference method (VQ) which constructs the ELBO via a quadratic lower bound to the likelihood terms [5]. Prediction is made by squashing the samples from the predictive distributions of the latent values at test points through the softmax likelihood for all methods.

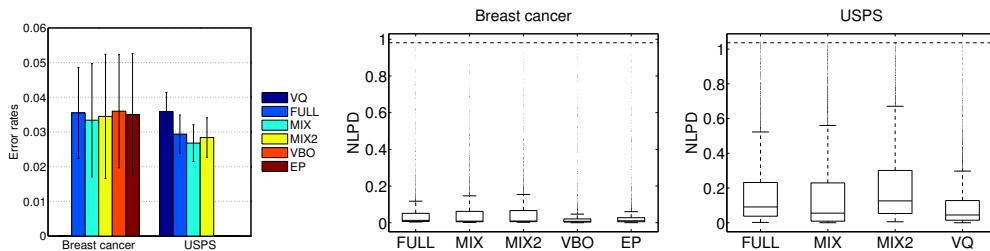

Figure 3: *Left plot*: classification error rates averaged over 5 runs (the error bars show two standard deviations). The AGP methods have classification errors comparable to the hard-coded implementations. *Middle and right plots*: the distribution of NLPD of all methods on the binary and multi-class classification tasks, respectively. The hard-coded methods are slightly better than AGP.

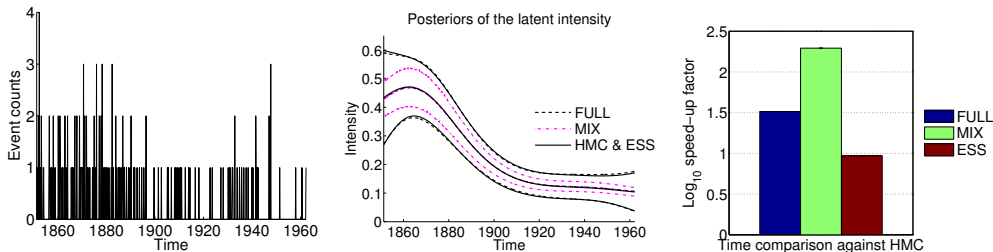

Figure 4: *Left plot*: the true event counts during the given time period. *Middle plot*: the posteriors (estimated intensities) inferred by all methods. For each method, the middle line is the posterior mean and the two remaining lines enclose 90% interval. AGP-FULL infers the same posterior as HMC and ESS while AGP-MIX obtains the same mean but underestimates the variance. *Right plot*: speed-up factors against the HMC method. The AGP methods run more than 2 orders of magnitude faster than the sampling methods.

The classification error rates and the NLPD are shown in Figure 3 for both tasks. For binary classification, the AGP methods give comparable performance to the hard-coded implementations, VBO and EP. The latter is often considered the best approximation method for this task [6]. Similar results can be observed for the multi-class classification problem.

We note that the running times of our methods are comparable to that of the hard-coded methods. For example, the average training times for VBO, EP, MIX, and FULL are 76s, 63s, 210s, and 480s respectively, on the Wisconsin dataset.

### 4.4   Log Gaussian Cox process (LGCP)

The LGCP is an inhomogeneous Poisson process with the log-intensity function being a shifted draw from a Gaussian process. Following [4], we use the likelihood $p(y_n|f_n) = \frac{\lambda_n^{y_n} \exp(-\lambda_n)}{y_n!}$, where $\lambda_n = \exp(f_n + m)$ is the mean of a Poisson distribution and $m$ is the offset to the log mean. The data concerns coal-mining disasters taken from a standard dataset for testing point processes [24]. The offset $m$ and the covariance hyperparameters are set to the same values as in [4].

We compare AGP with the Hybrid Monte Carlo (HMC, [25]) and elliptical slice sampling (ESS, [4]) methods, where the latter is designed specifically for GP models. We collected every 100th sample for a total of 10k samples after a burn-in period of 5k samples; the Gelman-Rubin potential scale reduction factors [26] are used to check for convergence. The middle plot of Figure 4 shows the posteriors learned by all methods. We see that the posterior by AGP-FULL is similar to that by HMC and ESS. AGP-MIX obtains the same posterior mean but it underestimates the variance. The right plot shows the speed-up factors of all methods against the slowest method HMC. The AGP methods run more than two orders of magnitude faster than HMC, thus confirming the computational advantages of our method to the sampling approaches. Training time was measured on a desktop with Intel(R) i7-2600 3.40GHz CPU with 8GB of RAM using Matlab R2012a.

## 5   Discussion

We have developed automated variational inference for Gaussian process models (AGP). AGP performs as well as the exact or hard-coded implementations when testing on five models using six real world datasets. AGP has the potential to be a powerful tool for GP practitioners and researchers when devising models for new or existing problems for which variational inference is not yet available. In the future we will address the scalability of AGP to deal with very large datasets.

### Acknowledgements

NICTA is funded by the Australian Government through the Department of Communications and the Australian Research Council through the ICT Centre of Excellence Program.

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
