[Supplementary Material]

# Automated Variational Inference for Gaussian Process Models
## *Supplementary Material*

**Trung V. Nguyen**
ANU & NICTA
VanTrung.Nguyen@nicta.com.au

**Edwin V. Bonilla**
The University of New South Wales
e.bonilla@unsw.edu.au

## 1   Proof of theorem 1

As discussed in the paper, the proof of theorem 1 directly follows from equation (7) and (8) in the main text whose detailed derivations follow.

$$\mathbb{E}_{q_k}[\log p(\mathbf{y}|\mathbf{f})] = \int q_k(\mathbf{f}|\boldsymbol{\lambda}_k) \log p(\mathbf{y}|\mathbf{f})\mathrm{d}\mathbf{f} \tag{1}$$

$$= \sum_{n=1}^{N} \int q_k(\mathbf{f}|\boldsymbol{\lambda}_k) \log p(\mathbf{y}_n|\mathbf{f}_{n\bullet})\mathrm{d}\mathbf{f} \tag{2}$$

$$= \sum_{n=1}^{N} \int q_{k(n)}(\mathbf{f}_{n\bullet}|\boldsymbol{\lambda}_{k(n)}) \log p(\mathbf{y}_n|\mathbf{f}_{n\bullet})\mathrm{d}\mathbf{f}_{n\bullet}, \tag{3}$$

$$= \sum_{n=1}^{N} \mathbb{E}_{q_{k(n)}} \log p(\mathbf{y}_n|\mathbf{f}_{n\bullet}) \tag{4}$$

where the last equality is an application of the following identity,

$$\int p(\mathbf{x}, \mathbf{z})h(\mathbf{x})\mathrm{d}\mathbf{x}\mathrm{d}\mathbf{z} = \int h(\mathbf{x})p(\mathbf{x}) \int p(\mathbf{z}|\mathbf{x})\mathrm{d}\mathbf{z}\mathrm{d}\mathbf{x} = \int p(\mathbf{x})h(\mathbf{x})\mathrm{d}\mathbf{x} \tag{5}$$

for any joint distribution $p(\mathbf{x}, \mathbf{z})$ and arbitrary function $h(\mathbf{x})$.

The gradients corresponding to $\mathbf{f}_{n\bullet}$ are thus given by

$$\nabla_{\boldsymbol{\lambda}_{k(n)}}\mathbb{E}_{q_k} \log p(\mathbf{y}|\mathbf{f}) = \nabla_{\boldsymbol{\lambda}_{k(n)}} \int q_{k(n)}(\mathbf{f}_{n\bullet}|\boldsymbol{\lambda}_{k(n)}) \log p(\mathbf{y}_n|\mathbf{f}_{n\bullet})\mathrm{d}\mathbf{f}_{n\bullet} \tag{6}$$

$$= \int \nabla_{\boldsymbol{\lambda}_{k(n)}} q_{k(n)}(\mathbf{f}_{n\bullet}|\boldsymbol{\lambda}_{k(n)}) \log p(\mathbf{y}_n|\mathbf{f}_{n\bullet})\mathrm{d}\mathbf{f}_{n\bullet} \tag{7}$$

$$= \int q_{k(n)}(\mathbf{f}_{n\bullet}|\boldsymbol{\lambda}_{k(n)})\nabla_{\boldsymbol{\lambda}_{k(n)}} \log q_{k(n)}(\mathbf{f}_{n\bullet}|\boldsymbol{\lambda}_{k(n)}) \log p(\mathbf{y}_n|\mathbf{f}_{n\bullet})\mathrm{d}\mathbf{f}_{n\bullet} \tag{8}$$

$$= \mathbb{E}_{q_{k(n)}}\nabla_{\boldsymbol{\lambda}_{k(n)}} \log q_{k(n)}(\mathbf{f}_{n\bullet}|\boldsymbol{\lambda}_{k(n)}) \log p(\mathbf{y}_n|\mathbf{f}_{n\bullet}) \tag{9}$$

$$\approx \frac{1}{S} \sum_{s=1}^{S} \nabla_{\boldsymbol{\lambda}_{k(n)}} \log q_{k(n)}(\mathbf{f}_{n\bullet}^s|\boldsymbol{\lambda}_{k(n)}) \log p(\mathbf{y}_n|\mathbf{f}_{n\bullet}^s), \tag{10}$$

where $\mathbf{f}_{n\bullet}^s \sim q_{k(n)}(\mathbf{f}_{n\bullet}|\boldsymbol{\lambda}_{k(n)})$. Here we have used the fact that $\nabla_{\mathbf{x}}f = f\nabla_{\mathbf{x}} \log f$ for any nonnegative function $f(\mathbf{x})$.

## 2 Proof of theorem 2

Since this proof is for the case of mixture with $K = 1$, we simply denote the posterior mean and variance as $\mathbf{m}$ and $\mathbf{S}$, respectively. From the expression of the ELBO,

$$\mathcal{L}(\mathbf{m}, \mathbf{S}) = \mathbb{E}_q[-\log q(\mathbf{f}|\boldsymbol{\lambda})] + \mathbb{E}_q[\log p(\mathbf{f})] + \mathbb{E}_q[\log p(\mathbf{y}|\mathbf{f})], \tag{11}$$

the gradients in (34), (35), and equation (4) we obtain the gradients of the ELBO w.r.t $\mathbf{S}$:

$$\nabla_{\mathbf{S}}\mathcal{L}(\mathbf{m}, \mathbf{S}) = \sum_{n=1}^{N} \nabla_{\mathbf{S}}\mathbb{E}_{q_{(n)}} \log p(\mathbf{y}_n|\mathbf{f}_{n\bullet}) + \frac{1}{2}\mathbf{S}^{-1} - \frac{1}{2}\mathbf{K}^{-1}. \tag{12}$$

Using the fact that $\mathbf{S}$ is a block diagonal covariance matrices and $q_{(n)}$ is a Gaussian with diagonal covariance we can take the gradients w.r.t $\mathbf{S}_j$, the covariances corresponding to the latent functions, which leads to

$$\nabla_{\mathbf{S}_j}\mathcal{L}(\mathbf{m}, \mathbf{S}) = \sum_{n=1}^{N} \nabla_{\mathbf{S}_j}\mathbb{E}_{q_{j(n)}} \log p(\mathbf{y}_n|\mathbf{f}_{n\bullet}) + \frac{1}{2}\mathbf{S}_j^{-1} - \frac{1}{2}\mathbf{K}_j^{-1}, \tag{13}$$

where $q_{j(n)}$ is the marginal posterior corresponding to the latent function $j$ and data point $n$. It is easy to see that the gradients of the likelihood terms w.r.t $\mathbf{S}_j$ is zero everywhere except on the diagonal. Denoting this diagonal as $\boldsymbol{\lambda}_j$ with elements $\lambda_{jn} = \nabla_{(\mathbf{S}_j)_{n,n}}\mathbb{E}_{q_{j(n)}} \log p(\mathbf{y}_n|\mathbf{f}_{n\bullet})$ we can rewrite the gradient as

$$\nabla_{\mathbf{S}_j}\mathcal{L}(\mathbf{m}, \mathbf{S}) = \boldsymbol{\Lambda}_j + \frac{1}{2}\mathbf{S}_j^{-1} - \frac{1}{2}\mathbf{K}_j^{-1}, \tag{14}$$

where $\boldsymbol{\Lambda}_j$ is the diagonal matrix with diagonal $\boldsymbol{\lambda}_j$. Setting the above to zero to derive the optimum condition we get,

$$\mathbf{S}_j = \left(\mathbf{K}_j^{-1} - 2\boldsymbol{\Lambda}_j\right)^{-1}. \tag{15}$$

This proves that the full covariance of the posterior can be parametrized with $\{\boldsymbol{\lambda}_j\}$, thus requiring only $\mathcal{O}(N)$ parameters for each latent function and hence $\mathcal{O}(QN)$ in total.

## 3 Proof of theorem 3

First we review Rao-Blackwellization [1], which is also known as partial averaging or conditional Monte Carlo. Suppose we want to estimate $V = \mathbb{E}[h(\mathbf{X}, \mathbf{Y})]$ where $(\mathbf{X}, \mathbf{Y})$ is a random variable with probability density $p(\mathbf{x}, \mathbf{y})$ and $h(\mathbf{X}, \mathbf{Y})$ is a random variable that is a function of $\mathbf{X}$ and $\mathbf{Y}$. It is easy to see that

$$\mathbb{E}[h(\mathbf{X}, \mathbf{Y})] = \int p(\mathbf{x}, \mathbf{y})h(\mathbf{x}, \mathbf{y})\mathrm{d}\mathbf{x}\mathrm{d}\mathbf{y} \tag{16}$$

$$= \int p(\mathbf{y})p(\mathbf{x}|\mathbf{y})h(\mathbf{x}, \mathbf{y})\mathrm{d}\mathbf{x}\mathrm{d}\mathbf{y} \tag{17}$$

$$= \mathbb{E}_{\mathbf{y}}[\underbrace{\mathbb{E}_{\mathbf{x}|\mathbf{y}}[h(\mathbf{X}, \mathbf{Y})|\mathbf{Y}]}_{\hat{h}(\mathbf{Y})}] \tag{18}$$

and, from the conditional variance formula,

$$\mathrm{var}[\hat{h}(\mathbf{Y})] < \mathrm{var}[h(\mathbf{X}, \mathbf{Y})]. \tag{19}$$

Therefore when $\hat{h}(\mathbf{Y})$ is easy to compute, it can be used to estimate $V$ with a lower variance than the original estimator. When $p(\mathbf{x}, \mathbf{y}) = p(\mathbf{x})p(\mathbf{y})$, the $\hat{h}(\mathbf{Y})$ is simplified to

$$\hat{h}(\mathbf{Y} = \mathbf{y}) = \int p(\mathbf{x})h(\mathbf{x}, \mathbf{y})\mathrm{d}\mathbf{x} = \mathbb{E}_{\mathbf{x}}[h(\mathbf{X}, \mathbf{Y})|\mathbf{Y}]. \tag{20}$$

We apply Rao-Blackwellization to our problem with $\mathbf{f}_{n\bullet}$ playing the role of the conditioning variable $\mathbf{Y}$ and $\mathbf{f}_{(-n)}$ playing the role of $\mathbf{X}$, where $\mathbf{f}_{(-n)}$ is $\mathbf{f}$ excluding $\mathbf{f}_{n\bullet}$.

First, we express the gradient of $\boldsymbol{\lambda}_{k(n)}$ as an expectation by interchanging the integral and gradient operators giving

$$\nabla_{\boldsymbol{\lambda}_{k(n)}} \mathbb{E}_{q_k} \log p(\mathbf{y}|\mathbf{f}) = \mathbb{E}_{q_k} \nabla_{\boldsymbol{\lambda}_{k(n)}} \log q_k(\mathbf{f}|\boldsymbol{\lambda}) \log p(\mathbf{y}|\mathbf{f}). \tag{21}$$

The Rao-Blackwellized estimator is thus

$$\hat{h}(\mathbf{f}_{n\bullet}) = \int q(\mathbf{f}_{(-n)}) \nabla_{\boldsymbol{\lambda}_{k(n)}} \log q_k(\mathbf{f}|\boldsymbol{\lambda}) \log p(\mathbf{y}|\mathbf{f}) \mathrm{d}\mathbf{f}_{(-n)} \tag{22}$$

$$= \int q(\mathbf{f}_{(-n)}) \nabla_{\boldsymbol{\lambda}_{k(n)}} \log q_{k(n)}(\mathbf{f}_{n\bullet}|\boldsymbol{\lambda}_{k(n)}) \log p(\mathbf{y}|\mathbf{f}) \mathrm{d}\mathbf{f}_{(-n)} \tag{23}$$

$$= \left( \nabla_{\boldsymbol{\lambda}_{k(n)}} \log q_{k(n)}(\mathbf{f}_{n\bullet}|\boldsymbol{\lambda}_{k(n)}) \right) \int q(\mathbf{f}_{(-n)}) \log p(\mathbf{y}_n|\mathbf{f}_{n\bullet}) \mathrm{d}\mathbf{f}_{(-n)} \tag{24}$$

$$+ \left( \nabla_{\boldsymbol{\lambda}_{k(n)}} \log q_{k(n)}(\mathbf{f}_{n\bullet}|\boldsymbol{\lambda}_{k(n)}) \right) \int q(\mathbf{f}_{(-n)}) \log p(\mathbf{y}_{-n}|\mathbf{f}_{(-n)}) \mathrm{d}\mathbf{f}_{(-n)} \tag{25}$$

$$= \left( \nabla_{\boldsymbol{\lambda}_{k(n)}} \log q_{k(n)}(\mathbf{f}_{n\bullet}|\boldsymbol{\lambda}_{k(n)}) \right) \left( \log p(\mathbf{y}_n|\mathbf{f}_{n\bullet}) + C \right), \tag{26}$$

where $C$ is a constant w.r.t $\mathbf{f}_{n\bullet}$. This gives the Rao-Blackwellized gradient,

$$\nabla_{\boldsymbol{\lambda}_{k(n)}} \mathbb{E}_{q_k} \log p(\mathbf{y}|\mathbf{f}) = \mathbb{E}_{q_{k(n)}} \hat{h}(\mathbf{f}_{n\bullet}) \tag{27}$$

$$= \mathbb{E}_{q_{k(n)}} \nabla_{\boldsymbol{\lambda}_{k(n)}} \log q_{k(n)}(\mathbf{f}_{n\bullet}|\boldsymbol{\lambda}_{k(n)}) \log p(\mathbf{y}_n|\mathbf{f}_{n\bullet}), \tag{28}$$

which is exactly the gradient we obtained in (9). To arrive at the last equality, we used the fact that $\mathbb{E}_q \nabla \log q = 0$ for any $q$.

**Remark**  This Rao-Blackwellization is only applicable to the case with diagonal covariance as it satisfies the independent condition (i.e. $p(\mathbf{x}, \mathbf{y}) = p(\mathbf{x})p(\mathbf{y})$).

# 4 Derivation of $\mathbb{E}_q[\log p(\mathbf{f})]$

The negative cross-entropy can be computed as:

$$\mathbb{E}_q[\log p(\mathbf{f})] = \sum_{k=1}^{K} \frac{1}{K} \int q_k(\mathbf{f}|\mathbf{m}_k, \mathbf{S}_k) \log p(\mathbf{f}) \mathrm{d}\mathbf{f} \tag{29}$$

$$= \sum_{k=1}^{K} \sum_{j=1}^{Q} \frac{1}{K} \int \mathcal{N}(\mathbf{f}_{\bullet j}; \mathbf{m}_{kj}, \mathbf{S}_{kj}) \log \mathcal{N}(\mathbf{f}_{\bullet j}; \mathbf{0}, \mathbf{K}_j) \mathrm{d}\mathbf{f}_{\bullet j} \tag{30}$$

$$= \sum_{k=1}^{K} \sum_{j=1}^{Q} \frac{1}{K} \left[ \log \mathcal{N}(\mathbf{m}_{kj}; \mathbf{0}, \mathbf{K}_j) - \frac{1}{2} \operatorname{tr}(\mathbf{K}_j^{-1}\mathbf{S}_{kj}) \right] \tag{31}$$

$$= -\frac{1}{2K} \sum_{k=1}^{K} \sum_{j=1}^{Q} \left[ N \log 2\pi + \log |\mathbf{K}_j| + \mathbf{m}_{kj}^T \mathbf{K}_j^{-1} \mathbf{m}_{kj} + \operatorname{tr}(\mathbf{K}_j^{-1}\mathbf{S}_{kj}) \right] \tag{32}$$

# 5    Gradients of $-\mathbf{KL}[q(\mathbf{f}|\boldsymbol{\lambda})||p(\mathbf{f})]$ w.r.t the variational parameters

Let $\mathcal{L}_{\text{ent}} = \mathbb{E}_q[-\log q(\mathbf{f}|\boldsymbol{\lambda})]$ and $\mathcal{L}_{\text{cross}} = \mathbb{E}_q[\log p(\mathbf{f})]$ then the gradients are given as following.

$$\nabla_{\mathbf{m}_k}\mathcal{L}_{\text{cross}} = -\frac{1}{K}\mathbf{K}^{-1}\mathbf{m}_k \tag{33}$$

$$\nabla_{\mathbf{m}_k}\mathcal{L}_{\text{ent}} = \frac{1}{K}\sum_{l=1}^{K}\frac{1}{K}\Big(\frac{\mathcal{N}_{kl}}{z_k} + \frac{\mathcal{N}_{kl}}{z_l}\Big)(\mathbf{S}_k + \mathbf{S}_l)^{-1}(\mathbf{m}_k - \mathbf{m}_l)$$

$$\nabla_{\mathbf{S}_k}\mathcal{L}_{\text{cross}} = -\frac{1}{2K}\mathbf{K}^{-1} \tag{34}$$

$$\nabla_{\mathbf{S}_k}\mathcal{L}_{\text{ent}} = \frac{1}{2K}\sum_{l=1}^{K}\frac{1}{K}\Big(\frac{\mathcal{N}_{kl}}{z_k} + \frac{\mathcal{N}_{kl}}{z_l}\Big)\Big[(\mathbf{S}_k + \mathbf{S}_l)^{-1} - (\mathbf{S}_k + \mathbf{S}_l)^{-1}(\mathbf{m}_k - \mathbf{m}_l)(\mathbf{m}_k - \mathbf{m}_l)^T(\mathbf{S}_k + \mathbf{S}_l)^{-1})\Big]$$

$$\tag{35}$$

where $\mathcal{N}_{kl} = \mathcal{N}(\mathbf{m}_k; \mathbf{m}_l, \mathbf{S}_k + \mathbf{S}_l)$ and $z_k = \sum_{l=1}^{K}\frac{1}{K}\mathcal{N}_{kl}$. Since the covariance matrices $\mathbf{S}_k$ have block structure, care should be taken in implementation such that its inversion can be done by inverting its blocks, i.e. the covariance corresponding to individual latent functions.

## 6    Predictive mean and variance by a mixture posterior

Using the mixture of Gaussians posterior, the predictive distribution for the new test points $\mathbf{x}_*$ can be approximated by

$$p(\mathbf{Y}_*|\mathbf{x}_*) = \frac{1}{K}\sum_{k=1}^{K}\underbrace{\int p(\mathbf{Y}_*|\mathbf{f}_*)\int p(\mathbf{f}_*|\mathbf{f})q_k(\mathbf{f})\mathrm{d}\mathbf{f}\mathrm{d}\mathbf{f}_*}_{p_k(\mathbf{Y}_*|\mathbf{x}_*)}, \tag{36}$$

where $p_k(\mathbf{Y}_*|\mathbf{x}_*)$ is the predictive distribution by component $k$. If this distribution has predictive mean $\mu_{k*}$ and variance $\sigma_{k*}^2$, the mean and variance of $p(\mathbf{Y}_*|\mathbf{x}_*)$ are given by:

$$\mathbb{E}[\mathbf{Y}_*] = \frac{1}{K}\sum_{k=1}^{K}\mu_{k*}, \tag{37}$$

$$\mathrm{Var}[\mathbf{Y}_*] = \frac{1}{K}\sum_{k=1}^{K}\sigma_{k*}^2 + \frac{1}{K}\sum_{k=1}^{K}\mu_{k*}^2 - \mathbb{E}[\mathbf{Y}_*]^2, \tag{38}$$

In other words, the predictive mean is the average of the prediction by each posterior component and the variance is the average of the variances by the components plus the variance of the mean prediction.

## References

[1] George Casella and Christian P. Robert.    Rao-Blackwellisation of sampling schemes. *Biometrika*, 1996.