[Reviews · NeurIPS 2014]

Submitted by Assigned_Reviewer_2

The authors present a flexible variational inference method geared Gaussian process models with various likelihoods. Specifically, they derive an inference method for models where some fixed number of latent functions (with GP priors that depend on the input covariate) parameterize a likelihood for conditionally independent observations.

They use variational inference to obtain the posterior over the latent functions, where the variational family of distributions is taken to be a mixture of Gaussians with some fixed number of components, and some covariance complexity (full, diagonal, block diagonal, etc). The paper derives the standard evidence lower bound (ELBO), which decomposes into a negative KL term and an expected log-likelihood term, and they note some convenient properties of these decompositions (re: optimizing covariance function parameters).

This paper is well written, very clear, and technically sound. The experiments section is done well and quite thorough - many likelihoods are tested and reported on.

Though using a mixture of Gaussians as a KL distribution is not particularly original, but the authors do develop a sampling technique to cope with the consequences of using a flexible KL distribution. This seems to be the main contribution of the paper - for latent Gaussian models they've derived an alternative to INLA and MCMC that has comparable performance and ostensibly computational advantages.

For impact, this method could be quite useful. The class of models it applies to is quite broad. It is unclear to me how much of an advantage it has over existing methods, such as nonparametric variational inference (Gershman, et al, 2012). I'd be interested in seeing a direct comparison.

Questions for the authors:
- What are the tradeoffs between computation and time/ease of optimizing the KL objective with respect to number of samples for the noisy gradients?
- Are there any consequences to using the lower bound of the KL term (between q(f|\lambda) and p(f)), which is a piece of the lower bound of the KL objective?
- Do you have any strategies for coping with larger samples, where the cubic inversion cost is high?
- Are there any comparisons between INLA and this method in low enough dimensions for INLA to be able to handle it? Are the results comparable? Comparisons with this method and nonparametric variational inference?
- Are 50k samples for HMC and ESS necessary? Is that a fair comparison to make time-wise?
Summary: I believe the paper is technically sound, very clearly written, and outlines a potentially very useful inference method for a wide class of models. I believe it would be a valuable contribution to NIPS.

Submitted by Assigned_Reviewer_12

I have read the authors feedback.

Summary of the Paper:

The paper describes a black-box variational inference method that is specific to models based on Gaussian processes. The method works by applying variational inference as the approximation method and by using a Gaussian mixture as the approximating distribution. The authors show that the lower bound in variational inference can be further lower bounded as the sum of a term that approximates the entropy of a Gaussian mixture and a cross-entropy term between the prior and the posterior approximation. A last term remains which is an expectation of the log likelihood of the model with respect to the posterior approximation. The nice thing is that the gradients of all terms except the one that depends on the likelihood can be computed in closed form. The gradient of the remaining term can be specified in terms of an expectation with respect to the posterior approximation and they can be stochastically approximated by using samples from a Gaussian distribution. In summary, the authors describe a method to carry out variational inference without the need to computed closed form expectations that may not be analytically tractable. The advantage of the method is that it computes gradients with respect to the hyper-parameters very easily. The method is compared in several experiments with deterministic VI and Markov chain Monte Carlo methods.

Clarity:

The paper is very well written and the use of the English language is correct. However, a few points in the paper should be further clarified. I am wondering what happens when the log of the likelihood cannot be computed. For example, this may happen when the likelihood factors are Heaviside functions of the latent functions. I also do not understand the paragraph between lines 251 and 253. It seems that the authors say that they need not use stochastic optimization methods. However, the do approximate stochastically the gradient of the expected log likelihood. I find this statement a bit contradictory. Another point is that the authors say that they iterate until the changes in the ELBO are less than 1e-5 or the changes in the variational parameters are smaller than 1e-3. Are they evaluating the ELBO always? I think their method can be applied when this is not possible.

Originality:

The method described can be understood as the application of the more general method described in [16] to the specific case of Gaussian process models. The method in [16] is more general and can be applied to a range of broader models. However, for this same reason, it does not exploit the particularities of Gaussian process models. Thus, I believe the work presented here is original.

Besides [16] the authors could also mention a recently published work. Namely,

Michalis K. Titsias and Miguel Lazaro-Gredilla. Doubly Stochastic Variational Bayes for non-Conjugate Inference. Proc of the 31st International Conference on Machine Learning (ICML 2014)

that is based on similar principles.

Quality:

I believe the quality of the paper is good in general and the experiments are relevant and representative for the method proposed. The same can be said about the methods the authors compare with. One thing that could be criticized is that only 5 runs are carried out and this might be insufficient to extract more conclusions from the experiments. Another thing that puzzles me is that there seems to be no clear advantage in the experiments for using a mixture of two components when compared a single component. This questions a bit the selection of that family for the posterior approximation.

Significance:

The experiments carried out in the paper show that the proposed method perform well in general and equivalently to other approaches based on Markov chain Monte Carlo or on deterministic variational inference. The advantage is that it needs not the computation in closed form of expectations with respect to the likelihood. These expectations are approximated by computing empirical averages with respect to a Gaussian distribution. The disadvantage is that this process is computationally more expensive, but not as expensive as Markov chain Monte Carlo methods. Again, the only point that questions a bit the significance of the proposed method is the fact that there seems to be no clear advantage of using a mixtures of Gaussians as the approximating distribution. An important point is that the authors claim advantages with respect to the method described in [16]. Specifically, they say they can very easily infer the kernel hyper-parameters, while with the method of [16] this is much more difficult. However, they do not compare with the particular approach of [16]. This questions the significance of the proposed method. A last criticism is that the authors only report gains in computation speed with respect to HMC in Section 4.4. I wonder what are the results when compared to VBO or EP. The method proposed is going to be more expensive, but by how much?

Strengths:

- Very well written manuscript.
- Theorems to support the claims of the paper.
- Simple but effective method.
- Many methods compared on very different settings.

Main Weaknesses:

- The use of a Gaussian mixture is not supported by the experiments.
- Only 5 folds are used in the experiments to compute averages.
- Not comparing with respect to related methods such as [16].
Summary: Interesting paper with nice results, but the lack of comparison with [16] questions the significance of the paper.

Submitted by Assigned_Reviewer_28

This paper presents a black-box variational inference procedure for supervised Gaussian process models. The idea is to save human time, as opposed to computer time, by reducing the amount of effort needed to perform approximate inference on a new GP model. This paper is very similar in spirit to that of Ranganath, Gerrish and Blei on black-box variational inference but it targets Gaussian process models. This allows for a more tailored black-box algorithm than in the generic case.

The variational distribution on the latent function values is imposed to be a mixture of Gaussians. Then, the evidence lower bound is split in its standard three terms. The first term is further lower bounded using the particular form of the variational distribution. The second term is analytically tractable. And the third term (which contains the likelihood) is approximated via Monte Carlo.

The article is well written and provides a clear motivation, an overview of the state of the art and an extensive set of experiments.

My main issue with this paper is the absence of sparse GPs to improve scalability. This is left for future work. I think that, as a community, we need to publish GP methods that are scalable from day one. Otherwise, we face an embarrassment when someone comes to us with a dataset that is too large. And in my experience this happens quite often! Unless we start demonstrating scalability on a regular basis, the field of Gaussian processes may become a niche in machine learning (even more than it currently is).

Miscellaneous comments:

- In more than one place there is mention of gradients of the parameters instead of gradient of the ELBO wrt the parameters.

- The right plot in Fig. 4 would be more easily readable if the y-axis tick labels were changed from 1, 2, 3… to 10, 100, 1000…
Summary: This is a well written paper presenting a variational back box inference procedure for supervised Gaussian process models. Although the problem tackled is interesting, there are not many particularly new insights and the very important issue of scalability is left for future work.
Author Feedback
Author rebuttal: We thank the reviewer for their comments.

Assigned_Reviewer_12

1. Comparison to [16] (BBVI)
We disagree with the statement “the method described can be understood as the application of the more general methods described in [16]” as our approach is different and radically better. First, [16] is specific to fully factorized variational distributions (see their equation 4). Our approach also considers the full Gaussian case and shows that the corresponding expectations (and gradients) can still be computed using samples from *univariate* Gaussians. Second, we exploit the decomposition of the ELBO to learn the hyperparameters using their exact gradients. We emphasize that [16] does not provide an analytical or practical way of learning hyperparameters and these are all set to fixed values (see [16], sec 5.5).

Nevertheless, as mentioned in lines 83-85, we had implemented [16] and extended it to estimate hyperparameters also using noisy gradients. We note that the cost of estimating these gradients is O(Ns x N^3) when using Ns samples from the posterior. Our method takes only O(N^3) as the exact expressions for the gradients are used. The results by BBVI are significantly worse. For example, on the housing dataset, BBVI gives a SSE mean of 0.30 compared to SSE of 0.11 by FULL. Similarly, BBVI gives a NLPD mean of 5.11 compared to NLPD of 4.36 by FULL. Furthermore, avg. training time of BBVI took 7 hours compared to 22 mins of FULL.

2. Using 5 runs
The distributions shown in the plots are over *all of the test points in 5 runs*. Even for the smallest test case (housing), a distribution has 1,030 samples. For the largest case (Abalone), a distribution has 15,885 samples. We believe such number of samples sufficiently represents the test distributions.

3. Advantage of mixture with 2 components
The mixture with 2 components does not seem to perform better than one component in the experiments in the paper but having the option of a multi-modal posterior is a contribution to the ML community. Remarkably, theorem 1 holds for all the distributions considered. Theorem 3, which is specific to the mixture (which is computationally cheaper than the full Gaussian) implies faster convergence during optimization.

4. Comparison to VBO/EP re computation
The key advantage of our method is that it minimizes model-specific derivations and implementations like VBO/EP. Time is more precious for humans than computers so we did not compare training time of our method to the hard-coded methods. However, here are some numbers for comparison. On the Breast dataset, average training times for VBO, EP, MIX, and FULL are 76s, 63s, 210s, and 480s respectively. On the Ionosphere dataset, average training times for VBO, EP, MIX, and FULL are 36s, 50s, 70s, and 300s, respectively.

5. Lines 251 and 253: stochastic gradients and stochastic optimization
The ELBO and its gradients are approximated via sampling but we treat them as ‘exact’ and use them with off-the-shell optimizers (e.g. CG or LBFGS). Since these methods can find the step sizes, we do not have to tune the learning rates unlike stochastic optimization.

6. Using the ELBO for checking convergence
In principle, we don’t need to compute the ELBO and can just use the noisy gradients with stochastic optimization. However, our implementation uses the ELBO and its gradients as inputs to off-the-shelf optimizers (see also 5 above).

Assigned_Reviewer_2

1. Tradeoffs between computation and time/ease of optimization wrt number of samples
Our approach uses samples from *univariate* Gaussians so a small number of samples is sufficient - for example, we used 2,000 in all experiments. We also experimented with 5,000 or 10,000 samples but it did not seem to affect the ease of convergence or quality of the approximations.

2. Consequences to using the lower bound of the KL term?
The lower bound seems to work well in our experiments (and in [7, 19] where it was also used) so we don’t think there are noticeable practical consequences to the optimization problem.

3. Strategies for coping with larger datasets?
See Assigned_Reviewer_28 item 1.

4. Comparison with INLA?
We compared with exact inference methods (for standard and warped regression), the best hard-coded inference methods (for classification), and MCMC which are all better than INLA.

5 Comparison with nonparametric variational inference (NPV)?
NPV is not a black-box method. It requires the first and second derivatives of the likelihood models. In contrast, our black-box approach only requires the evaluation of the likelihood.

6. How many samples to use with HMC and ESS?
We used 50,000 samples to avoid having to manually diagnose convergence of the samplers which is time consuming. Per the reviewer’s comment, we used the Gelman-Rubin potential scale reduction factors (PSRF) to check for convergence of the samples, which seems to suggest that 10,000 samples may be sufficient. We will update the paper to reflect this new finding. We note though that the running time of our method is still better than ESS and orders of magnitude faster than HMC.

Assigned_Reviewer_28

1. Scalability
All of the existing black-box methods (e.g. HMC, ESS) applicable to GP models suffer from scalability issues. Our approach significantly reduces the computational costs against method such as HMC and ESS, as demonstrated in the experiments. This is a contribution that should not be overlooked. As most GP models, our framework is amenable to low-rank approximations to the Gram matrix. For scalability to very large datasets, it might be possible to apply techniques such as ‘inducing points’ along with stochastic optimization. However, this is well beyond the scope of this paper.

2. Not many particularly new insights
We refer the reviewer to the three Theorems which all have made our approach practical.